



# The Italian Archive of Historical Earthquake Data, ASMI

Andrea Rovida[1,*], Mario Locati[1], Andrea Antonucci[1], Romano Camassi[2]

1) Istituto Nazionale di Geofisica e Vulcanologia, via A. Corti 12, 20133 Milano, Italy
2) Istituto Nazionale di Geofisica e Vulcanologia, viale Berti Pichat 6/2, 40127 Bologna, Italy

* *correspondence to*: Andrea Rovida (andrea.rovida@ingv.it)

**Abstract.** ASMI, the Italian Archive of Historical Earthquake Data, is a data collection distributed online that provides seismological data on more than 6600 earthquakes that occurred in the Italian peninsula and surrounding areas from 461 BC to the present day, based on more than 460 seismological data sources. ASMI is the Italian node of AHEAD, the European Archive

of Historical Earthquake Data, which is, in turn, the European node providing data on historical earthquakes to EPOS ERIC, the European Plate Observing System, a European Research Infrastructure Consortium. ASMI distributes earthquake parameters, sets of macroseismic intensity data and other details about earthquake effects, together with the bibliographical reference of the data source and, if possible the data source itself. ASMI's web portal allows users to query the data by earthquake or by data source, and to download the earthquake parameters and macroseismic intensities and represent them on interactive maps and

tables. ASMI is updated regularly with new data on past and recent earthquakes. ASMI is the basic source of data for the Italian Macroseismic Database (DBMI) and the Parametric Catalogue of Italian Earthquakes (CPTI). This article describes the archive content and structure, its main features and functionalities, and its potential seismological research applications.

## 1 Introduction

Detailed knowledge of long-term seismicity is fundamental for understanding a region's geodynamic and tectonic processes and the correlated natural hazards. Italy has a long tradition in the study of past earthquakes and has been very active and productive in the field of modern historical seismological research. The systematic collection of macroseismic data in Italy started in the late nineteenth century and complemented the previous collection of information about past earthquakes in seismological compilations, culminating with the work of Baratta (1901). After the termination of this collection with World War I, in the

1970s and 1980s the compilation of parametric catalogues, mostly at the regional scale, started. In the framework of the massive project on the geodynamics of Italy ("PFG - Progetto Finalizzato Geodinamica"), these catalogues merged into a national one (Postpischl, 1985a), with 81 strongest earthquakes jointly studied for the first time by seismologists and historians (Postpischl, 1985b). After 1985, a huge research programme aimed at improving the content of the PFG national catalogue, with the full involvement of historians. This programme focussed on both i) the revision of major earthquakes (e.g., Boschi et al., 1995; 1997)

complementing those analysed for locating nuclear power plants, and ii) the revision and assessment of macroseismic intensities of many events aimed at the compilation of an updated earthquake catalogue for the assessment of seismic hazard (Camassi and Stucchi, 1997). Both research groups produced the first ground-breaking digital macroseismic databases of the world, namely CFTI (Catalogue of Strong Italian Earthquakes, Boschi et al., 1995; 1997) and DOM (Monachesi and Stucchi, 1997). Afterwards, this activity continued less systematically, nonetheless updating the studies of many earthquakes, assessing macroseismic

intensities distributions for already known earthquakes, introducing unknown ones, and removing fake events. These investigations further improved the knowledge of Italian past seismicity which, coupled with both macroseismic data and instrumental recordings of recent earthquakes, is fully represented in the latest versions of the Italian parametric Earthquake Catalogue CPTI15 (Rovida et al., 2020a; 2022a) and the Italian Macroseismic Database DBMI15 (Locati et al., 2022), among the most advanced and complete databases of long-term seismicity in Europe.



Thanks to over 40 years of research, each earthquake in Italy is presently investigated in one or more historical seismological studies and provided with several successive updates of its macroseismic intensity data. The ensemble of these data constitutes an enormous patrimony that deserves to be preserved and made fully available to both present and future generations of researchers and to the general public. For this purpose, the Italian Archive of Historical Earthquake Data ASMI (Archivio Storico Macrosimico Italiano; Rovida et al., 2017) was conceived, developed, and made publicly accessible on the web. The starting

point was the internal inventory of earthquake data that the authors of the first versions of the Italian Parametric Earthquake Catalogue CPTI (namely CPTI99 and CPTI04; CPTI Working Group, 1999; 2004) established for the collection and comparison of the macroseismic datasets and catalogues, both historical and instrumental, to be used as input for that catalogue. As a result, ASMI is today a repository of data on more than 6600 Italian earthquakes between 461 B.C. and the present, derived from more than 460 data sources of different types. For each earthquake, various kinds of data are collected, organised, and made accessible

to provide the users with a wide perspective on the diversity of the available information and its development through different stages of research. As a result, ASMI collects and provides instrumental and macroseismic earthquake parameters from current and previous catalogues, together with their bibliographic metadata, related intensity data and additional information on earthquake effects.

This paper aims to describe the data contained in ASMI, their management, quality control, and ingestion in a relational database,

and to show the underlying IT infrastructure. It also illustrates the different ways of accessing data, i.e., the web portal and the web services, and the link with other data portals and databases.

## 2 Data description

### 2.1 Content of the archive

ASMI is a data collection that makes available the multiplicity of the existing information on the seismicity of the Italian territory

and surrounding areas by providing the most recent datasets published for each earthquake. Indeed, the peculiarity of ASMI is that, unlike seismic catalogues, it collects and presents multiple data sets for each earthquake. These datasets derive from any published source of information regarding earthquakes with effects in Italy, including historical investigations providing or not intensity data points, macroseismic intensity databases, macroseismic bulletins and the results of macroseismic surveys, and earthquake catalogues, including instrumental ones (Fig. 1). Considered data sources can be grouped into two broad categories:

i) macroseismic studies and ii) parametric catalogues (see also Rovida et al.; 2020b). The minimum information available for an earthquake from both descriptive studies and catalogues is the date of occurrence, which is reported for each event, and usually its area, of occurrence. In addition, ASMI reports earthquake parameters, such as epicentral coordinates and magnitude from parametric catalogues, and macroseismic intensity data points (MDPs) from macroseismic studies. MDPs consist of the name and coordinates of a place affected by a given earthquake and the macroseismic intensity value quantifying the effects of that

earthquake on people and buildings, according to a macroseismic scale. Parameters may also be supplied in macroseismic studies. In addition, macroseismic studies may provide descriptions of the earthquake effects without assessing any intensity data.

Depending on the source of data, ASMI collects and organizes all these different information, and provides access to them in different ways according to the selected access method (see Section 4).

As of today, ASMI in general contains and distributes data from:

● all the reference datasets used for compiling the different versions of the Italian Macroseismic Database (DBMI), from DBMI04 (Stucchi et al., 2007) to DBMI15 version 4.0 (Locati et al., 2022) and the Parametric Catalogue of Italian Earthquakes (CPTI), from CPTI99 (CPTI Working Group, 1999) to CPTI15 version 4.0 (Rovida et al., 2022a);



- all the reference datasets of the forerunners of DBMI and CPTI, namely the NT4.1 catalogue (Camassi and Stucchi; 1997) and the DOM macroseismic database (Monachesi and Stucchi, 1997);

- all the subsequent versions of the Italian Catalogue of Strong Earthquakes (CFTI), from the first version by Boschi et al. (1995) to the latest CFTI5med (Guidoboni et al., 2018; 20197);

- the macroseismic catalogue of the Mt Etna area (Catalogo Macrosismico dei Terremoti Etnei, CMTE; Azzaro and D'Amico, 2019);

- Historical seismological studies not considered in the catalogues and databases mentioned above;

- National catalogues and macroseismic datasets from neighbouring countries related to earthquakes that occurred at the Italian border (e.g., Fäh et al., 2011; SisFrance, 2016; Manchuel et al., 2018; Jomard et al., 2021).

In addition to all the earthquakes listed in the most recent version of CPTI, ASMI also considers those below their energy thresholds (intensity 5 and/or magnitude 4) if provided by recent sources. ASMI is updated whenever a new data source is published, and today it contains records from 460 data sources related to 6665 earthquakes that occurred from 461 B.C.

ASMI identifies the most reliable dataset among those available for each earthquake, carefully comparing and analysing them. Such an analysis considers each dataset's update level, the underlying research's quality and completeness, and the data's consistency and robustness. The data sources identified as the most reliable and representative of the most complete knowledge of each earthquake are selected as the reference input for each earthquake in the Italian Macroseismic Database DBMI and the Italian Parametric Earthquake Catalogue CPTI (Fig. 1).




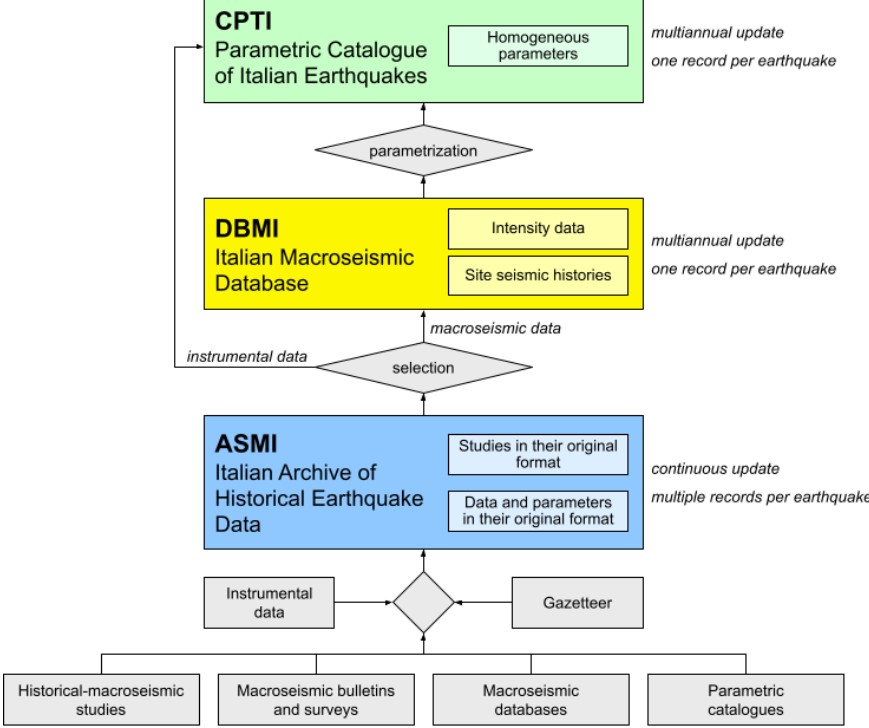

**Figure 1: Flowchart of the ASMI building blocks and their relation with DBMI and CPTI.**

### 2.2 Data ingestion, management and quality control

ASMI represents the Italian node of the European Archive of Historical Earthquake Data AHEAD (Albini et al. 2013; Locati et al. 2014; Rovida and Locati 2015) and shares with it the main structure and quality control of the input data. The latter includes data retrieval, processing and validation steps, and control points. The uploaded data are prepared, checked, and validated to ensure their conformity to ASMI's standards and formats. Inconsistencies in the data structure are identified through careful quality controls involving both relational database tools and manual revisions. The seven data ingestion, publication, and quality control steps are described below (Fig. 2).

1. Data source identification. New sources of earthquake data are identified in the literature through journals' mailing lists and social networks or contributed by the authors or the ASMI collaborators.

2. Data source examination and validation. The data intended for inclusion in ASMI must provide earthquake parameters, macroseismic data points, or textual descriptions of earthquake effects. Publicly accessible data sources in any media (journal articles, conference proceedings, public reports, online databases, etc.) and the licence associated with the data sources are considered. The first control point consists of verifying the fulfillment of ASMI's objectives and requirements and the possibility of redistributing both the original data source and the provided data.

3. Compilation of bibliographic metadata. The source of data is registered and its standard bibliographic metadata are compiled and stored in the relational database.

4. Extraction and standardisation of event data. Event data (e.g., date, location, magnitude) are identified within the data source and are then standardised (e.g. date and/or geographical coordinates conversion). A new record of the dedicated event table of the relational database is manually compiled for each event.

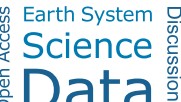

5. Attribution of the event identifier. Through the analysis of the parameters and the additional information contained in the data source, each new record of the events table, representing an event, is associated with an event identifier, either already existing or newly created in case a non-existing event is introduced. The second control point verifies the consistency of all earthquake parameters and the association with other earthquakes in ASMI or the uniqueness of new ones through the event identifier.

6. Extraction and standardisation of intensity data. In case the data source also provides macroseismic intensity data, they are analysed and standardised, and a record with each observation is manually added to the corresponding table of the ASMI relational database. Each place mentioned in the data source is georeferenced using a Gazetteer, an Italian-wide unified reference system internally developed over the years with unique place identifiers, names, and geographical coordinates. Once referred to the Gazetteer, Intensity values are homogenised (e.g., Roman to Arabic numerals). For each intensity point, both the original and the homogenised place names, coordinates, intensity values, and their macroseismic scale are stored in the table. The third control point aims to check the consistency of the intensity distribution in both the table and the map.

7. Data publication. The data source and its metadata, the event(s) data, and the related intensity distribution(s) are published online after the last control point, verifying the event's correct visualisation and the association's correctness with other data sources. Once the data and metadata are compiled, verified, and made compliant with the FAIR principles (Findability, Accessibility, Interoperability, Reusability; Wilkinson et al., 2016), they are made openly available on ASMI.

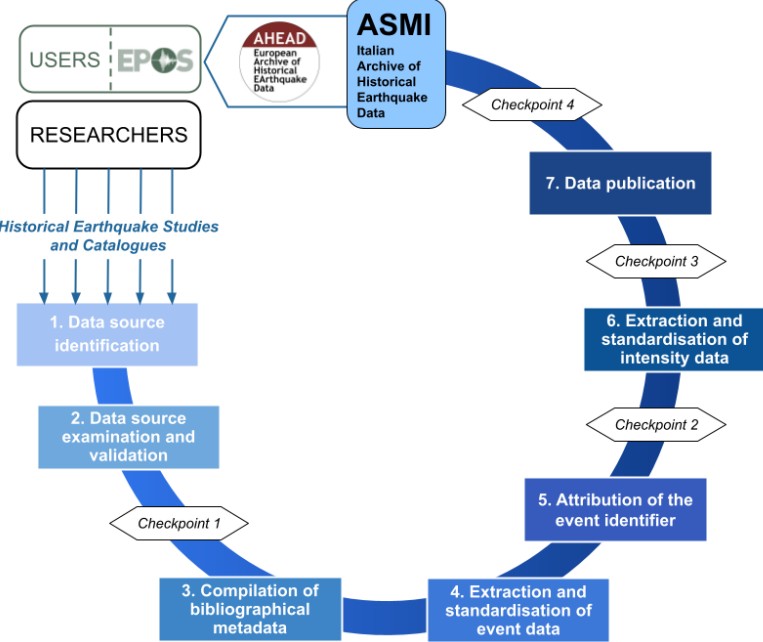

**Figure 2: Workflow for the data quality assurance in ASMI.**

**2.3 Relational database structure**

ASMI and AHEAD are quite similar in terms of data types, therefore they share a very similar relational database (Locati et al., 2014). The database schema design (Fig. 3) allows for efficient storage, retrieval, and analysis of data, and it also ensures data integrity and consistency by separating each piece of information into thematic tables connected using foreign keys to link related



elements. Below is a description of the most relevant tables of the database, following as much as possible the order of the data ingestion procedure.

a) Data Sources. This table contains entries describing the considered data sources. The table includes details such as the title, the author(s), the container (i.e., journal, book, database) where the data source is published, the year of publication, and, if available, the link to a web page hosting the data source, possibly through its DOI code. Each data source is assigned a unique identification code ("data source ID").

b) Earthquake data. This table contains an entry for each earthquake mentioned in each data source (identified through its assigned "record ID"). Together with the identification code of the data source ("data source ID") of origin, each entry reports the date of the earthquake, the epicentral area, the geographical coordinates of the epicentre, the epicentral intensity, the estimated magnitude(s) and their scale (i.e., Mw, ML, Ms, mb), the original earthquake identifier, if present, and the available macroseismic intensity data in terms of the identifier of the MDP set ("MDP set ID"). An identifier is assigned to each earthquake ("event ID") to group entries related to the same earthquake. Finally, the table "List of earthquakes" is dynamically generated to uniquely list all the "event ID" contained in the table "Earthquake data".

c) MDP, Macroseismic Data Points. This table contains an entry for each macroseismic intensity datapoint, to which a unique "MDP ID" is assigned. The same identifier ("MDP set ID") is assigned to all macroseismic intensities related to the same earthquake and derived from a given data source. This table contains information related to the macroseismic intensity data of a given earthquake, such as the attributes of the affected place (name and geographical coordinates) and the assessed intensity value. The original data reported in the data source are also retained in the table (see Section 2.2). For each place with an assigned intensity value, the corresponding "place ID" (see below) is reported. The list of MDP sets is dynamically created in a dedicated table ("MDP sets") with its maximum observed intensity, total number of MDPs and macroseismic scale.

A series of additional tables are also present, below is a brief description of them:

a) Authors. This table lists all the authors of the archived data sources identified by an "author ID", which allows to query data sources on their author(s).

b) Earthquake information. This table contains an overall description of the event and earthquake effects other than macroseismic intensities that may be reported in the data source, such as fatalities, injuries, and seismogeological effects.

c) MDP information. This table contains descriptive information about the effects related to a specific MDP.

d) Gazetteer. This table contains a gazetteer where each locality has a place name and geographical coordinates. Additional place names are also stored, such as old ones, or exonyms in multiple languages. Every locality is associated with a unique identifier ("place ID") and with the administrative subdivisions it belongs to, which, according to ISTAT, the Italian National Institute of Statistics, have four levels. Each level's information is stored in a separate table, all linked through their respective identifiers (Admin. subdivisions 0, 1, 2 and 3).

e) Rosetta events. The table contains the association of ASMI's "event ID" of each earthquake with the unique identifiers of the same earthquake in other databases for interoperability's sake (see Section 5).

f) Registered users and comments. An additional table stores the credentials of registered users, i.e., data curators and contributors, who have access to data that are not publicly available and can add comments on specific events stored in the relevant table.



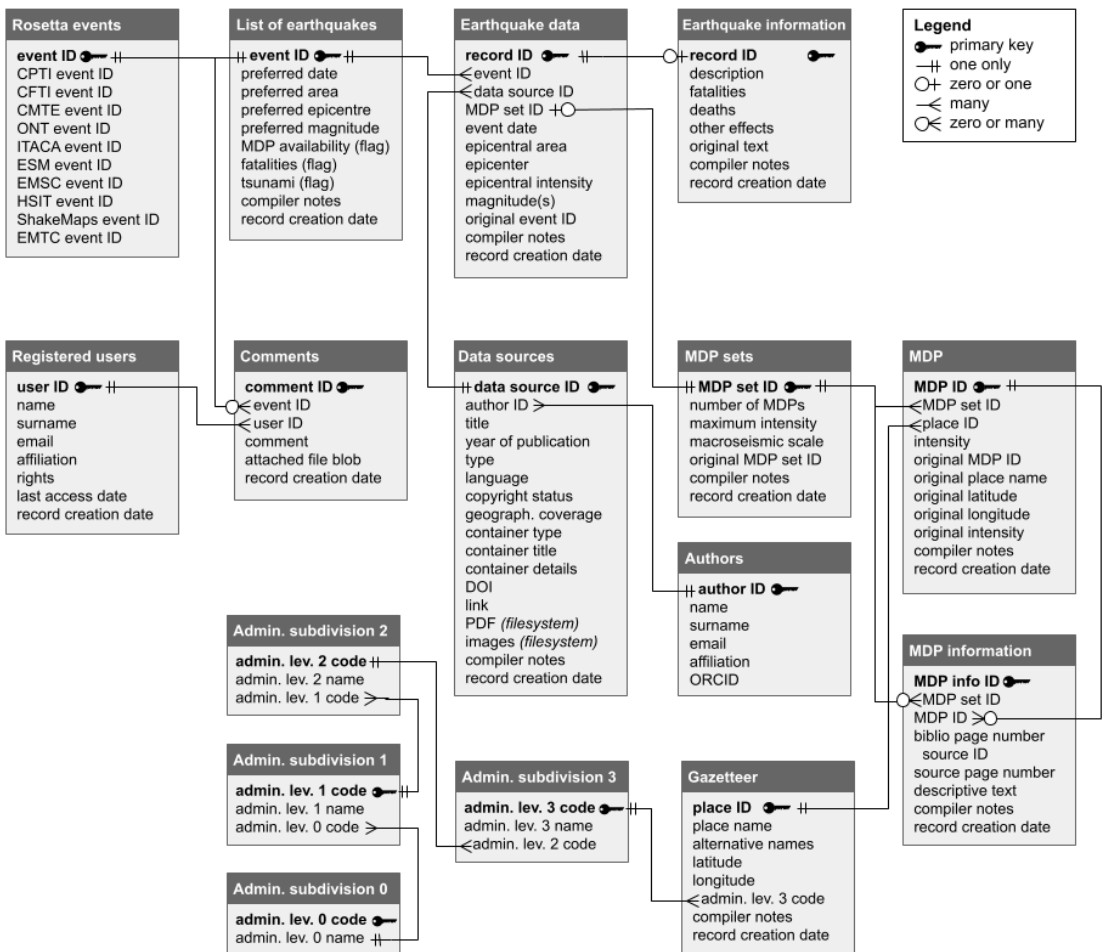

**180**  **Figure 3: Simplified schema of ASMI's relational database, explaining the relation among data tables and their primary key; the type of relation between table keys is also represented.**

## 3 IT infrastructure

The IT infrastructure underlying ASMI is named "Emidius" - the saint protecting people against earthquakes according to the Catholic tradition - and is focused on disseminating long-term seismological data. In particular, its core relational database and

**185**  metadata management are designed for the preservation, inventorying, and public access of archived historical earthquake data.

### 3.1 The underlying "Emidius" infrastructure

The Emidius infrastructure hosts both ASMI and AHEAD together with other resources, such as all versions of CPTI and DBMI, the Global Earthquake History Archive and Catalogue (Albini et al., 2014), MIDOP, the Macroseismic Intensity Data Online Publisher software (Locati and Cassera, 2010) and BOXER (Gasperini et al., 1999; Gasperini et al., 2010), a tool to compute

**190**  focal parameters of earthquakes from macroseismic data.

To ensure the highest reliability of the web services for accessing data, the Emidius infrastructure is made of two physical nodes hosted in INGV data centres, the master one in Milan, and the backup in Bologna, and both are directly managed by INGV



personnel. The technical solution for managing the failover between the two nodes is described in Nannipieri et al. (2019). Both server nodes adopt the Proxmox Virtual Environment, an open-source virtualisation platform running two distinct Kernel-based

Virtual Machines (KVM), one for the web portals and RESTful-based web services, and another one for the Database Management Systems (DBMS) and the OGC (Open GeoSpatial Consortium) web services. In addition to the failover system between the two nodes, a backup on a NAS (Network-attached storage) is performed daily at the main node in Milan.

The adopted web server software is NGINX (https://nginx.org/), all web pages and most data services are written in PHP, whereas the services compliant with the Open GeoSpatial Consortium (OGC) standard are based on the open-source Geoserver

(https://geoserver.org/). The main DBMS is MySQL, whereas an instance of PostgreSQL with the PostGIS geographical extension is used as the data source for the Geoserver software. The double DBMS in place only partially overlap, MySQL is used to manage the entire ASMI and AHEAD archive content, whereas a snapshot of it is periodically copied in PostgreSQL to feed the web services for machine-friendly access to data.

Data curators directly manipulate data on the MySQL server at the main node in Milan, mostly using Microsoft Access software

via an ODBC (Open Database Connectivity) API (Application Programming Interface). In addition to a direct connection to the MySQL database, data curators may use a web-based and user-friendly graphical user interface with restricted access that facilitates the association of descriptive metadata to each input data source. The maintenance of the overall data infrastructure and the management of the database structure is supervised by a Data Manager who may operate as a Data Curator at the same time.

Although ASMI and AHEAD share most of the underlying IT infrastructure and the archived data partially overlap, ASMI differs in many aspects from AHEAD (Locati et al. 2014) regarding technological implementation and overall data policy management. ASMI is a dynamic national data infrastructure fully managed by INGV. In contrast, AHEAD is a more rigid data infrastructure covering Europe, directly managed by INGV but governed by 16 partner European organisations. AHEAD is based on regional nodes providing earthquake data for their respective area. Many tools offered by the graphical user interface of the web portal

are usually developed and first implemented in ASMI and then, after extensive testing, become available in AHEAD. Examples are i) the tool for comparing macroseismic intensity data sets from different data sources, ii) reference geographical layers from external data sources that users may add on top of the geographical maps, iii) the visualisation of additional information such as fatalities and environmental effects, or iv) the possibility to upload temporary macroseismic intensities from other earthquakes for comparing effects of various earthquakes on the same map.

**3.2 Metadata management**

Metadata management is relevant in ASMI as it enhances its usability, reliability, and value for final users. By dealing with metadata according to the FAIR data principles, ASMI facilitates machine-friendly access to its data. For example, the extensive use of DOI (Digital Object Identifiers) to identify elements managed in ASMI is important because each DOI is coupled with a set of metadata following well-defined schemas (e.g., Crossref, DataCite) that can be used to fine-tune search queries.

ASMI itself is associated with a DataCite DOI - https://doi.org/10.13127/asmi - assigned by INGV, and the related metadata (https://data.ingv.it/dataset/67) is managed in the INGV Data Registry (INGV Data Management Office, 2020). ASMI and its web services are described using the ISO 19115/19139 metadata standard to conform with the European INSPIRE Directive, the Infrastructure for Spatial Information in the European Community. ASMI is also described as a data trusted repository in re3data (https://www.re3data.org/), a Registry of Research Data Repositories and an additional DOI is associated with this descriptive

record (http://doi.org/10.17616/r31njnlh).





Particular care was devoted to enriching web pages of the "Query by Data Source" of the web portal to describe each seismological study carefully. In particular, the "head" element of the HTML source code includes metadata following the Dublin Core -both elements and terms-, the HighWire Press and the Open Graph standards. Given the amount of metadata included in these web pages, there is a high chance that users land on ASMI when searching for one of the considered data sources through the major web search engines (i.e., Google, Bing).

Among the available FAIR self-assessment models, guidelines, and tools, we selected the following solutions to evaluate ASMI compliance with the FAIR principles:

- Data and metadata management guidelines are provided by EPOS ERIC, the European Plate Observing System (Bailo et al, 2022; 2023a; 2023b). Being compliant with these guidelines is a requirement as ASMI indirectly provides data to EPOS via AHEAD (see 4.2);
- FAIR Indicators provided by the FAIR Data Maturity Model Working Group (2020; Bahim et al., 2020). These indicators were considered because they are developed by a well-established Research Data Alliance (RDA) Working Group and provide a comprehensive set of elements to consider for improving the overall FAIRness of the adopted technological solutions which cover both data and metadata management;

F-UJI automated FAIR data assessment open-source tool (Devaraju and Huber, 2020), developed in the framework of the Fostering FAIR Data Practices in Europe (FAIRsFAIR), a European Horizon 2020 project. By entering the DOI, the tool automatically provides a quantitative and qualitative evaluation with a detailed output explaining what was good and what can be improved on every check it performs. The tool is currently being developed, and we provide feedback to developers.

## 4 Data access

Data archived in ASMI are accessible in several ways, such as through a dedicated web portal (https://emidius.mi.ingv.it/ASMI/), web services (https://emidius.mi.ingv.it/ASMI/services/), via the AHEAD (https://www.emidius.eu/AHEAD/) and EPOS (https://www.ics-c.epos-eu.org/) data platforms, and with the "QQuake" plug-in for QGIS (Locati et al., 2021).

### 4.1 Web Portal

The ASMI web portal (https://emidius.mi.ingv.it/ASMI/) allows querying data either by earthquake or by data source. Through the "query by earthquake", it is possible to select a single earthquake from a list or a map (Fig. 4). The selection can be performed with several filters, i.e. by year or time span, number of intensity data, magnitude classes, and availability of information on tsunamis or casualties. An earthquake can be selected directly from a map, by clicking the symbol of the epicentre or by defining a circular or polygonal area, whose vertices can be saved, shared, and reused to reproduce the polygon at any time.



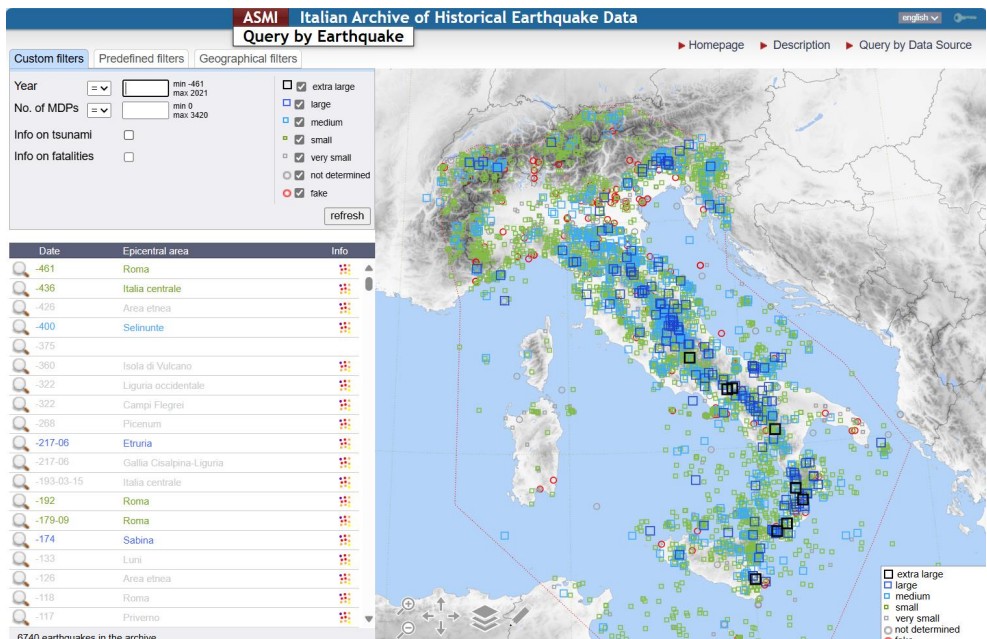

**Figure 4: Query by earthquake page of ASMI's web portal (https://emidius.mi.ingv.it/ASMI/query_event/; last accessed 12/11/2024).**

Once an earthquake is selected, all available information is displayed in a pop-up window presenting several tabs (Fig. 5). The tab "Catalogues" (Fig. 5a) shows the parameters of the selected event as defined in the latest version of CPTI (as of today CPTI15 v.4.0; Rovida et al., 2022a) and many other alternative and/or previous catalogues with their bibliographical reference. Each epicentre can be plotted on the map to show the possible differences in the estimates proposed by different catalogues and the parameters from all catalogues can be downloaded QuakeML format.

The tab "Studies" (Fig. 5b) lists all the archived seismological studies that deal with the selected earthquake. The bibliographic reference of each study is shown together with the available information about fatalities, the number of macroseismic intensity data, and the maximum observed intensity with the macroseismic scale used. The study contributing data to DBMI is highlighted and shown by default on the map. The related macroseismic intensity distribution can be shown on the map selecting the number of intensity data of a given study. MDP sets from single studies can be downloaded as text, QuakeML or Google Earth files. It is possible to compare the information contained in different studies with two dedicated tools. The "Compare MDP sets" tool lets the user compare all the macroseismic intensity distributions available for the same event through the list of the localities investigated by different studies with their estimates of macroseismic intensity. Alternatively, the tool "Add another study for comparison" allows users to add macroseismic data from one or more studies archived in ASMI to the map related to a different earthquake. Users can also temporarily upload their intensity distributions to compare them with existing datasets.

The tab "Seismicity" (Fig. 5c) shows the map of the epicentres and a time-magnitude chart of the earthquakes that occurred within an editable geographic area around the selected earthquake.

For 79 historical earthquakes with magnitude ≥ 6 that occurred between 1117 and 1968 C.E. the maps of ground shaking are shown in the "Shakemaps" tab (Fig. 5d) and are all downloadable in pdf format. As detailed in Oliveti et al. (2023a; 2023b), the shakemaps are calculated starting from the intensity data contributing to DBMI15 in terms of macroseismic intensity and five ground motion parameters (i.e., PGA, PGV, SA 0.3, SA 1.0, and SA 3.0) with the USGS-ShakeMap code (Michelini et al. 2020; Worden et al., 2020).



The "Tsunami" tab shows (Fig. 5e) the description, bibliography, and parameters related to 52 earthquakes with tsunami

information as described in the Euro-Mediterranean Tsunami Catalogue (EMTCv2; Maramai et al. 2014; 2019).

Finally, the "Other data" tab (Fig. 5f) provides information for earthquakes that occurred in the instrumental era for which a link
to the ITalian ACcelerometric Archive (ITACA v4.0; D'Amico et al., 2021; Felicetta et al. 2023) and the Osservatorio Nazionale
dei Terremoti (ONT; https://terremoti.ingv.it/) is provided.



**Figure 5: Tabs showing information and data available in ASMI for the earthquakes selected through the "Query by earthquake" of
the web portal: a) Catalogues, b) Studies, c) Seismicity, d) ShakeMaps, e) Tsunami, f) Instrumental data from ONT and ITACA
(https://emidius.mi.ingv.it/ASMI/query_event/; last accessed 12/11/2024).**



Users may add to the maps a series of layers from external services, for example, the latest version of DISS, the Database of Individual Seismogenic Sources (DISS Working Group, 2021) or the geographical boundary of the EPOS Near Fault Observatories (Chiaraluce et al., 2022).

The "Query by data source" is possible through the list of archived data sources (Fig. 6a). The data sources are listed in alphabetical order and each of them is displayed with its short citation and internal ID, complete reference, number of earthquakes
considered in ASMI, and related macroseismic data points, if available. The list of data sources may be filtered by author or by year of publication. The selection of a data source gives access to a pop-up window (Fig 6b) that displays its bibliographical information and abstract, together with a list and a map of the earthquakes from the study considered in ASMI. These, in turn, give access to the same event window described above. Most data sources are available as downloadable PDF files or through a link to an external webpage. Downloadable PDF files either consist of the entire seismological study or its parts related to single
earthquakes when they are complete and freestanding from the point of view of the earthquake information. On the contrary, the link to the data source's webpage is provided for papers published in copyrighted journals or books, and the possibility of downloading only their abstract is available.

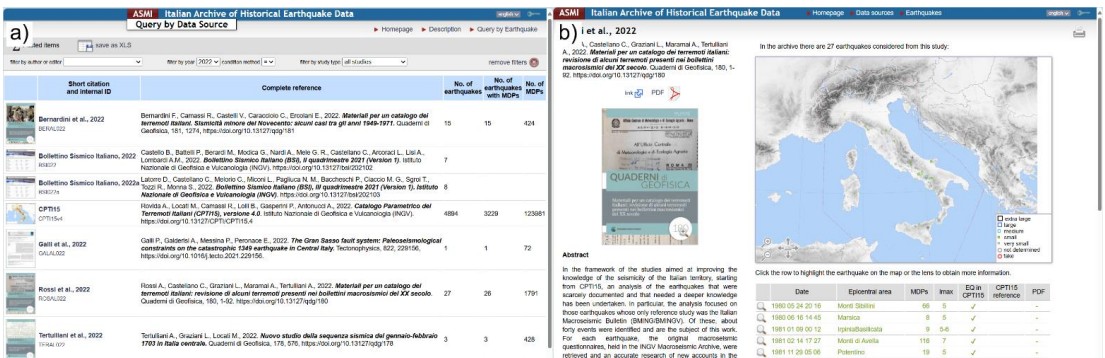

**Figure 6: a) Query by data source page of ASMI's web portal. b) Pop-up window showing the details of the selected data source, together with the list and the map of the earthquakes considered in ASMI (https://emidius.mi.ingv.it/ASMI/query_study/; last accessed 12/11/2024).**

### 4.2 Web Services

An alternative way of directly accessing ASMI data is through multiple web services, one for each data type archived (i.e., parameters, intensities, bibliography), as summarised in Table 1.

Earthquake parameters can be accessed using three distinct API standards: OGC (Open Geospatial Consortium; https://www.ogc.org/) WFS (Web Feature Service), OGC WMS (Web Map Service), and FDSN-event (International Federation of Digital Seismograph Networks; http://www.fdsn.org/webservices/):

● The OGC WFS standard is well established and supported by major GIS systems (e.g., QGIS), and is designed for transferring geographical vector features associated with data. As any other web service compliant with the OGS standards, it allows on-the-fly geographical reprojection to any requested coordinate system and it supports the CQL (Common Query Language) for filtering the data. Multiple data encodings are supported, such as GML (Geographic Markup Language), KML (Keyhole Markup Language), Shapefile, GeoJSON, CSV, and Microsoft Excel XLSX.



● The OGC WMS (Web Map Service) service provides georeferenced tiled raster maps in several output formats (Tab. 1) with a predefined built-in symbology style.

● The FDSN-event web service allows querying a catalogue of earthquakes following a standard provided by the FDSN. The service offers two distinct outputs: a simplified and compact output encoded in ASCII (CSV) format, which provides a reduced amount of information, and a complete output in the XML format, encoded using QuakeML v1.2 (Schorlemmer et

al., 2011). The latter format provides additional information about the data provenance and can associate multiple locations and multiple magnitudes estimates for each event in addition to the predefined one (i.e., that from CPTI15), allowing extended access to the entire data collection. In addition, an output is available in the GeoJSON encoding format, a non-FDSN-standard format that was specifically developed in the framework of the collaboration between AHEAD and the European-Mediterranean Seismological Centre (EMSC-CSEM; https://www.emsc.eu/) to provide earthquake parameters to

the EPOS Data Portal (https://www.ics-c.epos-eu.org/).

Macroseismic intensity data can be accessed using a RESTful web service. This service is based on the specifications of the FDSN web service for events, but additional and custom parameters have been added to meet the specificity of macroseismic intensity data. When these custom parameters are passed to provide macroseismic intensity data, the service switches from QuakeML v1.2 to QuakeML v2.0 output, enabling the macroseismic package (Locati, 2014). Additional query parameters allow,

for example, filtering by the number of available intensities, by the minimum intensity reported or by requesting intensity data from a specific data source among those archived in ASMI.

The bibliography service is developed to retrieve the bibliographic metadata of data sources and can be accessed using a dedicated RESTful service. This service offers the possibility of querying the data by author, title, year of publication, DOI identifier, and other bibliographic parameters. In addition, a geographic search is possible, providing all studies about

earthquakes whose epicentre falls in the user-defined area. The service works with the same logic as the fdsn-event web service and allows data encoding in XML format using the Dublin Core standard, RDF format, and plain text (Tab. 1).

A dedicated section of the ASMI web portal (https://emidius.mi.ingv.it/ASMI/services/)provides descriptions and of usage examples for the web services mentioned above.

| | Standard | Output | Output Formats |
|---|---|---|---|
| *Earthquake Parameters* | OGC WFS | Geographical features of CPTI15 origins and magnitudes | GML, KML, Shapefile, GeoJSON, CSV, XLSX |
| | OGC WMS | Styled maps with CPTI15 origins and magnitudes | PNG, JPG, GIF, PDF, GeoTiff |
| | FDSN event | CPTI15 origins and magnitudes and all alternative parameters from ASMI | QuakeML v1.2, GeoJSON, CSV |
| *Macroseismic Intensity Data* | RESTful | Set of macroseismic intensity data from a specific data source | QuakeML v2.0, GeoJSON, CSV |
| *Bibliography* | RESTful | Data source by author, title, year of publication, DOI identifier, and other bibliographic parameters | XML, RDF, CSV |

Table 1: List web services standards for accessing ASMI data.

**4.3 Relation to AHEAD and the EPOS data platforms**

ASMI represents the Italian node of the European Archive of Historical Earthquake Data AHEAD, therefore it provides data for all earthquakes that occurred in Italy in the time window 1000-1899. AHEAD collects earthquake catalogues and macroseismic intensities for the whole European continent, acting as a pan-European, common, and open platform supporting research in the





field of historical seismology, in the framework of a Memorandum of Understanding among 16 European organisations that run

regional nodes. Thanks to the shared underlying Emidius IT infrastructure, AHEAD provides the same web portal functionalities and web services APIs as those available for accessing ASMI.

Thanks to its coordination role at a European level, AHEAD is the data and service provider for seismological data related to pre-instrumental earthquakes in the framework of EPOS ERIC. In particular, AHEAD is a member of the EPOS Seismological Thematic Core Service (Haslinger et al., 2022) and coordinates its data management activities with organisations such as the

European-Mediterranean Seismological Centre (EMSC-CSEM; https://www.emsc.eu/), Observatories & Research Facilities for European Seismology (ORFEUS; https://www.orfeus-eu.org/) and European Facilities for Earthquake Hazard and Risk (EFEHR; http://www.efehr.org/). An overview of the multidisciplinary EPOS Delivery Frameworks at the European level is provided by Bailo et al. (2023a).

### 4.4 QQuake, a QGIS plug-in

QQuake, a QGIS plug-in (Locati et al., 2021) was developed using the Python language to provide easy access to ASMI data when working with the QGIS software. Both earthquake parameters from the considered catalogues and macroseismic intensity data can be retrieved easily by interacting with the graphical user interface of QQuake. Once downloaded, data is plotted on the map adopting a default symbology for a ready-to-use experience. QQuake allows the user to filter the data, for example, based on the magnitude, or selecting a particular geographical area or period. QQuake demonstrates how easily anyone can develop

third-party software by taking advantage of standardised services for accessing data, a real-world example of what can be achieved by adopting the FAIR data principles, in its turn, a required step for the adoption of the Open Science paradigm for a more efficient sharing of research outputs.

In addition to data from ASMI, QQuake provides access to a variety of other seismological resources, such as data from AHEAD, the European Preinstrumental Earthquake Catalogue (EPICA; Rovida and Antonucci, 2021; Rovida et al., 2022b), the Database

of Individual Seismogenic Sources (DISS; DISS Working Group, 2021), the Euro-Mediterranean Tsunami Catalogue (EMTC; Maramai et al., 2014), or the Italian Tsunami Effects Database (ITED; Maramai et al., 2021).

### 5 Data interoperability

Earthquake identifiers stored in ASMI (see Section 2.3 and Fig. 3) and the diffusion of web services for disseminating seismological data make ASMI interoperable with several online seismological databases. Each earthquake and each data source

archived in ASMI has a simplified web address URL specifically designed to pinpoint each element. Interoperability is ensured through the table "Rosetta events" of ASMI's relational database which contains the event identifiers of the same earthquake in diverse databases. Currently, translation among earthquake identifiers of different resources is used to enable internal ASMI functionalities in the web portal.

ASMI is interoperable with the Catalogue of Strong Earthquakes in Italy (CFTI5med; Guidoboni et al. 2018; 2019), which

provides detailed information on the historical sources and the effects of 1230 significant Italian earthquakes. The link to the corresponding earthquake information available in CFTI5Med is provided in the "Studies" tab of ASMI (see Section 4.1). Conversely, the web interface of CFTI5Med offers a direct link to the ASMI's corresponding earthquake webpage for common events.  ASMI is also connected with the historical catalogue of the Mt. Etna earthquakes (CMTE; Azzaro and D'Amico, 2019) for the events of that area.  The tab "Studies" of ASMI's web portal gives the full CMTE description of the event as from CMTE

(i.e., a summary of the earthquake effects along with details on the earthquake sequence and seismogeological effects), available for 38 events considered in ASMI.





ASMI is interoperable with the Euro-Mediterranean Tsunami Catalogue (EMTCv2; Maramai et al. 2014; 2019), whose data on earthquake-induced tsunamis are displayed in the relevant tab of the earthquake webpage of ASMI. In its turn, EMTCv2 shows earthquake parameters from the CPTI15 catalogue using ASMI's web services.

Web services also allow the mutual link of events between ASMI and the ITalian ACcelerometric Archive (ITACA; Felicetta et al., 2023) and the European Strong Motion Database (ESM; Luzi et al., 2020).

Finally, the Macroseismic Photographic Database (Database Fotografico Macrosismico, DFM; QUEST, 2022) interacts with ASMI's web services to retrieve the earthquake parameters to be associated with the field photographs of earthquake damage taken during macroseismic field surveys.

A further extension of the capability to translate among identifiers of additional data types is planned, for example, to translate identifiers of localities associated with macroseismic intensities and those published by ISTAT, the Italian National Institute of Statistics, or to link accelerometric recordings and macroseismic observations. Another planned activity to increase interoperability with ASMI is the publication of a new web service allowing users to translate event identifiers among different earthquake catalogues and online databases, such a service will be based on the "Rosetta events" translation table (Fig. 3). This 405 new service is part of a bigger set of planned services (Locati et al., 2016) taking advantage of machine-friendly and machine-actionable identifiers, metadata, and services improving the interoperability among data archives in the field of geosciences able to perform activities that were previously performed manually.

## 6 Data availability

ASMI full data collection (Rovida et al., 2017; https://doi.org/10.13127/asmi) is available through mulitple web service 410 (https://emidius.mi.ingv.it/ASMI/services/) as described in Section 4.2 and Table 1. The collection is distributed under a Creative Commons Attribution 4.0 International (CC BY 4.0) licence, and metadata are available at https://commons.datacite.org/doi.org/10.13127/asmi and https://data.ingv.it/dataset/67.

## 7 Conclusions

ASMI is a fundamental resource for advancing the understanding of seismic activity taking advantage of the huge scientific and 415 cultural Italian heritage, preserving the relevant documentation. ASMI provides a complete and fully accessible picture of the knowledge of Italian historical seismicity, from the earthquake locations and magnitudes assessed with the most modern seismic networks to the macroseismic effects of earthquakes that occurred in antiquity. The quality and reliability of the provided data are ensured by the data management and ingestion procedures, developed and refined during decades of experience in handling historical earthquake data. The original concept of the database, dedicated to the compilation of updated parametric catalogues, 420 has been expanded to incorporate and spread different types of data and information to various users, including historians, seismologists, geologists, geophysicists, teachers, and the general public.

The database structure and the consultation facilities are designed to incorporate newly published data continuously. The preservation and the thorough use of identifiers in the original datasets already allowed interoperability with other infrastructures providing seismological data, which can be expanded in the future. The extensive use of metadata, modern technologies for 425 accessing data, and an Open Data compliant license (CC BY 4.0) fully support the Open Science paradigm and the FAIR data principles.



**Author contributions**

All authors wrote the paper, AR, ML, and AA prepared the figures. All authors contribute to the ASMI archive content. ML works on IT infrastructures, web portal, and web services.

**Competing interests**

One of the authors is a member of the editorial board of Earth System Science Data.

**Acknowledgements**

We thank Massimiliano Stucchi and Paola Albini, formerly working at INGV, who greatly contributed to the shaping of the original ASMI concept and the overall design of its structure. We also thank all seismologists who more or less directly support
ASMI by publishing their research findings and macroseismic data on past earthquakes and providing their feedback as ASMI's users. In particular, we thank our colleagues Raffaele Azzaro, Filippo Bernardini, Beatriz Brizuela, Carlos Hector Caracciolo, Viviana Castelli, Salvatore D'Amico, Emanuela Ercolani, Laura Graziani, Alessandra Maramai, Vera Pessina, Antonio Rossi, Andrea Tertulliani and the authors of CFTI, the Catalogue of strong earthquakes in Italy and the Mediterranean area, in particular Gabriele Tarabusi and Cecilia Ciuccarelli. ASMI benefited from funding provided by the Italian Civil Protection Department
(DPC) as part of Annex B2, Work Package 1, Task 1 activities of the DPC-INGV Agreement 2019-2021; however, ASMI does not necessarily reflect the position and the official policies of the DPC. ASMI also benefits from funding provided by the Joint Research Unit of EPOS Italia.

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
