# Peer review of "The Italian Archive of Historical Earthquake Data, ASMI"

_Earth System Science Data, 2024_

## Author Response (AR1)

**AUTHORS' REPLY ON RC1**

**We sincerely thank the reviewer for the useful comment as well as for the kind words and the appreciation of our work.**

**Here follows (in bold characters) a point-by-point reply to RC1**

1a/It seems to me that a few comparative elements could be included in this paper to clarify the role of the DBMI database in relation to the ASMI database, which seems to include all the macro-seismic data present in DBMI. In my reading, DBMI seems to me to be a bit like a duplicate of ASMI (without the instrumental) in the Italian environment. I have to admit that it's not all that clear to me. What is the place/role of DBMI in relation to ASMI? Are there any data or services in DBMI that are not contained in or accessible through ASMI?

The relationship between ASMI and DBMI is shown in Figure 1 and described in lines 90-94. ASMI collects and presents all the datasets, especially macroseismic ones, available for an earthquake, whereas DBMI presents a single macroseismic dataset for each earthquake. This dataset is retained to be the most complete, updated, and thorough one and the most representative of the knowledge of the earthquake. The scope of DBMI is thus to provide a single intensity distribution for each event, from which the macroseismic parameters of CPTI are assessed. In addition, DBMI provides the timeseries of the earthquake effects recorded at any Italian locality (the so-called "seismic histories" of localities), accessible at: https://emidius.mi.ingv.it/CPTI15-DBMI15/query\_place/

**We will better clarify this in the revision of the manuscript.**

1b/No information is given on the quality and/or uncertainty associated with the MDP values and locations (cf Sisfrance A,B,C quality : safe, medium safe, uncertain). It would be interesting to explain the reasons for this (perhaps in 2.3, c/ MDP macroseismic datapoint (line 155)). For scientific use of the data, the qualitative value of the information can have a major impact on the results particularly for historical earthquakes.

We agree with the reviewer but very few data sources provide quality of the MDPs, and they use very different criteria to assess and present it. This discontinuity and inhomogeneity prevented us from publishing this information, which should be carefully explained to the users. Qualities associated with MDPs are stored in the database together with all the available information if provided by the data source.

We will better specify this in point 6 of the list in Section 2.2.

1c/In the ASMI database, polygons of uncertainty in the location of the macroseismic epicentre appear; it would also be interesting to specify this in the article.

**Unfortunately, no uncertainty in the locations is presented in ASMI.**

2/ line 69: ... 'the macroseismic intensity value quantifying the effects of that earthquake on people and buildings,...'

**Add also the effects on objects and furniture (used for low intensities in all macroseismic scales)**

**Correct, we thank the reviewer for pointing this out.**

3/ Line 150: ... 'epicentral intensity' specify whether this is a single calculation ASMI method for all events (if so, add a reference), or whether it is specific to each MDP dataset.

Data are archived in ASMI as presented in the data source with no elaborations, apart from the standardizations required by the database structure described in Section 2.2. We will emphasize this concept also in Section 2.3

4/ Line 161: ... 'with its maximum observed intensity' specify whether this is the maximum observed 'In Italy' or 'for the event whatever the country' or 'for the series of data displayed'.

As indicated, the "MDP sets" table is dynamically created and the maximum intensity is calculated as the maximum value within the dataset.

5/I think it would also be interesting for the reader to see the feedback or monitoring method used to track the use of this database by users, perhaps over the last year (national and international users, graphs of download, user feedback).

**We thank the reviewer for the suggestion. We will add a short paragraph and a figure to illustrate the access statistics.**

6/ in your conclusion, Perhaps you could write 2 or 3 lines detailing the future developments envisaged for the project, to place it in a dynamic evolutionary process, for example with regard to data visualisation tools, export formats, possible additional information on the data (epicentral distance of MDP), cross-border exchanges, possible revisions...

The Conclusions already mention future developments in terms of both i) continuous update with newly published data and ii) expansion of the interoperability with other seismological databases (lines 415-417). We will expand this part and integrate it with the description of the foreseen developments in data interoperability presently described at the end of Section 5 (lines 398-405).

Cross-border exchanges are conducted in the framework of the AHEAD initiative within EPOS, as extensively described in Section 4.3. Elaborations on the data are not foreseen within ASMI, which is a collection of existing and published data.

**AUTHORS' REPLY ON RC2**

We sincerely thank the reviewer for the nice comments on our work.

We will take into account and fix the small bugs he/she pointed out in the thorough review.